# Hypersensitivity reactions to multiple anti-tuberculosis drugs

**Hong-Joon Shin**[1], **Jin-Sun Chang**[1], **Min-Suk Kim**[1], **Bo-Gun Koh**[1], **Ha-Young Park**[1], **Tae-Ok Kim**[1], **Chul-Kyu Park**[1], **In-Jae Oh**[1], **Yu-Il Kim**[1], **Sung-Chul Lim**[1], **Young-Chul Kim**[1], **Young-Il Koh**[2], **Yong-Soo Kwon**[1]*

1 Division of Pulmonary and Critical Care Medicine, Department of Internal Medicine, Chonnam National University Hospital, Gwangju, South Korea, 2 Division of Allergy, Asthma, and Clinical Immunology, Department of Internal Medicine, Chonnam National University Hospital, Gwangju, South Korea

☯ These authors contributed equally to this work.

* yskwon@jnu.ac.kr

**Data Availability Statement:** All relevant data are within the manuscript.

**Funding:** This study was supported by a grant (BCRI20013) of Chonnam National University Hospital Biomedical Research Institute. The

## Abstract

### Objective

This study aimed to evaluate hypersensitivity reactions to anti-tuberculosis (TB) drugs.

### Methods

We retrospectively compared the clinical manifestations and treatment outcomes of single and multiple drug hypersensitivity reactions (DHRs).

### Results

Twenty-eight patients were diagnosed with anti-TB DHRs using oral drug provocation tests. Of these 28 patients, 17 patients (60.7%) had DHRs to a single drug and 11 (39.3%) had multiple DHRs. The median age of patients was 57.5 years (interquartile range [IQR], 39.2–73.2). Of the total patients, 18 patients (64.3%) were men. The median number of anti-TB drugs causing multiple DHRs was 2.0 (IQR 2.0–3.0). Rifampin was the most common drug that caused DHRs in both the single and multiple DHR groups (n = 8 [47.1%] and n = 9 [52.9%], respectively). The treatment success rate was lower in the multiple DHR group than in the single DHR group; however, the difference was not statistically significant (81.8% vs. 94.1%; P = 0.543).

### Conclusions

Multiple anti-TB DHRs were common in all patients who experienced DHRs, and rifampin was the most common causative drug. The treatment outcomes appeared to be poorer in patients with multiple DHRs than in those with single DHRs.

funders had no role in study design, data collection
and analysis, decision to publish, or preparation of
the manuscript.

**Competing interests:** NO authors have competing
interests

## Introduction

The standard treatment for drug-susceptible tuberculosis (TB) consists of treatment with iso-niazid, rifampin, ethambutol, and pyrazinamide for 2 months, followed by treatment with iso-niazid and rifampin for 4 months [1]. The treatment is highly efficacious, achieving cure rates of approximately 90%–95% [2–4]. However, adverse drug reactions (ADRs) to anti-TB drugs can cause significant morbidity, leading the discontinuation of these effective anti-TB drugs and worse outcomes. The risk of developing an ADR to anti-TB drugs has been variously reported from 8 to 85% depending on the population the studies enrolled and the severity of ADRs the studies evaluated [5].

A drug hypersensitivity reaction (DHR) is an important adverse drug reaction(ADR) defined as "objectively reproducible symptoms or signs initiated by exposure to a defined stimulus at a dose tolerated by normal persons" by the World Health Organization (WHO). A drug reaction with demonstrated immunological mechanisms, either antibody or cell mediated, is referred to as drug allergy [6]. DHRs can be classified as immediate or non-immediate DHRs depending on the onset time after drug exposure. Immediate DHRs typi-cally occur within 1–6 h after drug exposure, whereas non-immediate DHRs commonly occur at any time after 1 h of drug administration [7]. The degree of clinical presentations of DHRs varies, from mild (e.g., urticaria) to severe (e.g., drug reaction with eosinophilia and systemic symptoms [DRESS] syndrome or Stevens–Johnson syndrome [SJS]/toxic epi-dermal necrolysis [TEN]) [8, 9].

ADRs, including DHRs to highly effective first-line anti-TB drugs, are important because they may limit the use of these drugs or increase lost to follow-up rate, treatment failure, and relapse [10–13]. Moreover, if DHRs to multiple drugs occur [14], TB treatment can be a challenge owing to a significant lack of effective and tolerable anti-TB drugs. In our previ-ous study, peripheral eosinophilia during anti-TB therapy was common (17.8%) and cuta-neous ADRs were common in these patients with eosinophilia. However, we did not evaluate DHRs to multiple drugs in these patients [15]. Moreover, only a few case reports have evaluated the clinical features and treatment outcomes associated with DHRs to multi-ple anti-TB drugs [16–20]. Therefore, this study analyzed the clinical characteristics and treatment outcomes in patients with TB who experienced DHRs to single and multiple anti-TB drugs during treatment.

## Methods

### Patients

We retrospectively reviewed the clinical data of patients who underwent drug challenge tests owing to suspected DHRs while receiving anti-TB drugs, at Chonnam National Uni-versity Hospital between January 2011 and April 2016. Drug hypersensitivity was sus-pected in patients with skin rash and/or fever, anaphylaxis, and angioedema [21]. The diagnosis of TB based on either of the following: identification of *Mycobacterium tuber-culosis* by culture or polymerase chain reaction in clinical specimens or clinical, radiolog-ical, or histological findings consistent with TB and clinical responses to anti-TB treatment.

The Institutional Review Board of Chonnam National University Hospital (Gwangju, Republic of Korea) approved the study protocol and provided permission for reviewing and publishing this study, including information obtained from patient records (CNUH-01018-158). The requirement of informed consent was waived owing to the retrospective nature of the study, and patient information was anonymized and deidentified prior to analysis.

## Definition and classification of DHRs

We collected the data of patients who were confirmed to have DHRs to anti-TB drugs using the oral provocation test.

All patients were hospitalized for the oral provocation test when they had no symptoms related to DHRs; an allergologist and a trained nurse performed the test and monitored DHR-related symptoms and signs such as pruritus and skin rash. When the patient developed sign and symptoms of DHRs, the test was considered positive. The oral provocation test was performed in the order of pyrazinamide, isoniazid, ethambutol, and rifampin. However, according to the judgment of the clinician, the order of anti-TB drugs for oral provocation test may be changed in the order of drugs with the lowest probability of hypersensitivity. The test was conducted using escalating doses of each drug until attainment of a therapeutic dose, which was different from therapeutic dose we previously used [22]. The following dose escalation method was used: the patient's usual daily dose was set at 100% and dissolved in 100 mL of water for the challenge. Patients suspected to have an immediate immune response were orally administered volume of 1, 3, 10, and 56 mL solution at 1-h intervals, and the symptoms were observed. Patients suspected to have a delayed immune response underwent a drug challenge every 2 days. To perform the drug challenge, 10 and 30 mL of dissolved drug suspension were administered orally every 12h intervals and the reaction was observed 12 h after final dose administration.

Single and multiple DHR were defined as hypersensitivity reactions elicited by one and two or more chemically distinct drugs, respectively.

We classified multiple DHRs into two subtypes, simultaneous and sequential multiple DHRs, according to Gex-Collet et al [23]. Simultaneous and sequential multiple DHRs were defined as the occurrence of hypersensitivity to different administered drugs simultaneously and at time intervals, respectively.

An anaphyaxis was defined as an acute onset of an illness involving the skin, mucosal tissue, or both in more than one body system after exposure to a trigger [24].

SJS and TEN represent a spectrum of the most severe DHRs, wherein large areas of the epidermis become rapidly necrotic, resulting in skin detachment. SJS and TEN involve <10% and >30% of total body surface area, respectively [25].

DRESS syndrome was diagnosed based on the RegiSCAR criteria and required a minimum of three of the following eight clinical features: 1) fever ($\geq$38.5˚C); 2) lymphadenopathy; 3) eosinophilia; 4) atypical lymphocytes; 5) skin rash; 6) organ involvement; 7) resolution ($\geq$ 15 days); and 8) exclusion of other potential causes [26].

## Treatments of tuberculosis and management of DHRs

All patients with TB treated at our hospital were registered in the TB registry and were monitored for drug compliance and ADR occurrence during treatment by specially trained nurses who participated in the Public–Private Mix project for TB control in South Korea [27]. Anti-TB drugs were discontinued in patients with suspected DHR. Based on the decision of the attending physician, a combination of second-line anti-TB drugs was initiated or anti-TB drugs were withdrawn during drug challenge tests.

## Laboratory data

We collected laboratory parameters related to anti-TB treatment initiation (initial values) and DHR occurrence (peak values). The laboratory parameters included white blood cell, lymphocyte, and eosinophil counts and percentages as well as aspartate aminotransferase (AST) and alanine aminotransferase (ALT) levels.

### Treatment outcomes

For analyses, we used the WHO definitions for cure, treatment completion, treatment failure, death, loss to follow-up, and transfer out [1]. Treatment success was indicated by the favorable outcomes of cure and treatment completion. Treatment failure, death, and loss to follow-up were considered unfavorable outcomes.

### Statistical analysis

Data are presented as medians (interquartile range [IQR]) or numbers (percentages). Demographic and clinical variables were compared between the single DHR and multiple DHR groups using the chi-squared test for categorical variables or the Mann-Whitney U test for continuous variables. The Wilcoxon signed rank test was used to compare initial and peak laboratory variables within groups. Statistical analyses were performed using SPSS version 23.0 (IBM, Armonk, NY, USA), with a p value of $< 0.05$ considered statistically significant.

## Results

### Patient characteristics

During the study period, a total of 2,347 patients with TB received anti-TB drugs and 28 patients were diagnosed with DHRs to anti-TB drugs using the oral provocation test (Fig 1). Of these patients, two and eight patients underwent skin prick and patch tests, respectively, before the oral provocation test. One patient underwent both skin prick and patch test before the oral provocation test.

Of the 28 patients, one patient underwent an oral provocation test while switching to second-line anti-TB drugs. The remaining 27 patients underwent oral provocation tests after discontinuing all anti-TB drugs. After identifying the causative drugs for DHRs, 19 and nine patients were administered modified anti-TB regimens during hospitalization and in outpatient clinics, respectively (S1 Table).

Of the 28 patients, 17 (60.7%) had single DHRs and 11 (39.3%) multiple DHRs (Table 1). There were no differences between the two groups with respect to demographic characteristics

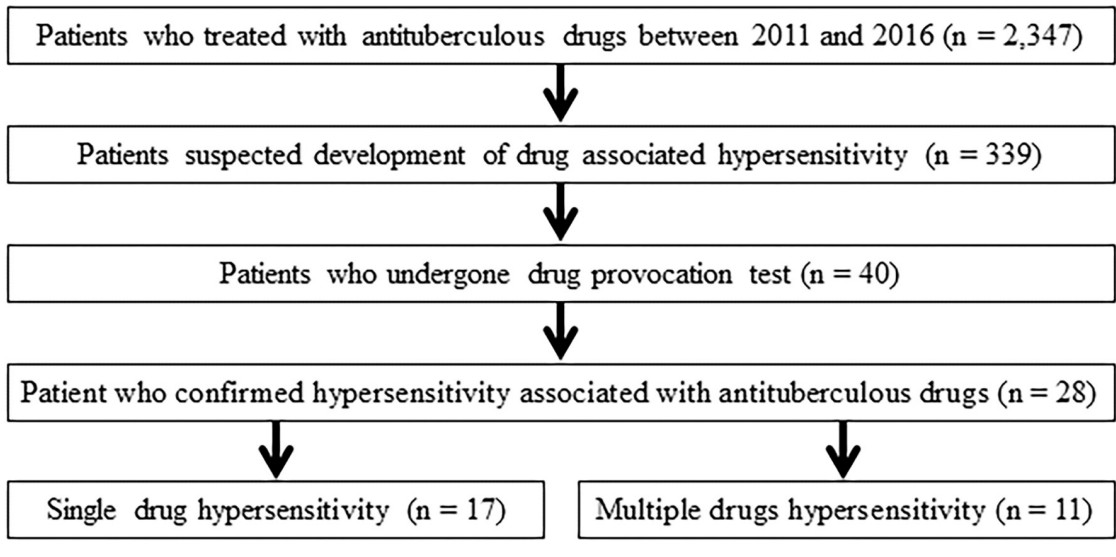

**Fig 1. Study flowchart.**

**Table 1. Baseline characteristics of patients in the single and multiple DHR groups.**

| Variables | Total (n = 28) | Single DHR (n = 17) | Multiple DHR (n = 11) | P value |
|---|---|---|---|---|
| Age, years | 57.5 (39.2–73.2) | 61.0 (50.5–72.5) | 44.0 (36.0–74.0) | 0.259 |
| Male sex | 18 (64.3%) | 10 | 8 | 0.689 |
| Body mass index | 22.1 (19.6–22.1) | 22.4 (20.5–25.1) | 21.6 (18.8–24.8) | 0.353 |
| Diabetes mellitus | 4 (14.3%) | 4 | 0 | 0.132 |
| Hypertension | 10 (35.7%) | 8 | 2 | 0.226 |
| Malignancy | 4 (14.3%) | 3 | 1 | 1.000 |
| Allergic disease | 4 (14.3%) | 2 | 2 | 0.628 |
| Site of tuberculosis | | | | 0.611 |
| Pulmonary only | 11 (39.3%) | 6 | 5 | |
| Extra-pulmonary only | 12 (42.9%) | 7 | 5 | |
| Combined pulmonary and extra-pulmonary | 5 (17.9%) | 4 | 1 | |
| Previous treatment history | 1 (3.6%) | 1 | 0 | 1.000 |
| Prior loss to follow-up | 3 (10.7%) | 1 | 2 | 0.543 |
| Clinical manifestations of drug hypersensitivity | | | | |
| Pruritus | 12 (42.9%) | 10 | 2 | 0.054 |
| Fever | 6 (21.4%) | 2 (11.8%) | 4 (36.4%) | 0.174 |
| Phenotype of DHR | | | | |
| Maculopapular exanthema | 27 (96.4%) | 16 | 11 | 1.000 |
| Anaphylaxis | 3 (10.7%) | 1 | 2 | 0.543 |
| SJS/TEN | 0 | 0 | 0 | 1.000 |
| DRESS syndrome | 3 (14.3%) | 1 | 2 | 0.543 |
| Interval between TB treatment initiation and first signs of DHR, days | 12.0 (7.0–43.2) | 14.0 (9.0–42.5) | 11.0 (6.0–65.0) | 0.572 |
| Interval between TB treatment discontinuation and drug challenge test, days | 13.5 (7.0–21.7) | 14.0 (9.0–21.5) | 11.0 (3.0–27.0) | 0.300 |
| Interval between TB treatment initiation and drug challenge test, days | 67.0 (34.2–129.7) | 70.0 (31.5–123.5) | 64.0 (35.0–157.0) | 0.817 |
| Interval between first signs of DHR and drug challenge test, days | 51.5 (25.2–82.2) | 47.0 (21.5–85.5) | 59.0 (28.0–75.0) | 0.572 |
| Inflammatory markers at the time of drug challenge test | | | | |
| WBC (×10³/mm³) | 4.9 (4.0–7.8) | 5.1 (4.0–8.0) | 4.8 (4.0–7.4) | 0.706 |
| Neutrophils (×10³/mm³) | 2.8 (1.7–4.4) | 3.0 (2.2–4.5) | 2.2 (1.2–4.2) | 0.290 |
| CRP (mg/dL) | 0.3 (0.1–0.9) | 0.2 (0.1–0.9) | 0.5 (0.1–1.9) | 0.437 |

Data are presented as number (percentage) or median (interquartile range). DHR, drug hypersensitivity reaction; TB, tuberculosis; WBC, white blood cell count; CRP, C-reactive protein

or the presence of underlying diseases. In both groups, most patients had never been treated for TB (94.1% in the single DHR group and 81.8% in the multiple DHR group). The most common signs of hypersensitivity reactions were skin rash (96.4%) and pruritus (42.9%), and there was no significant difference in clinical symptoms and signs between two groups. In the multiple DHR group, all patients were found to have simultaneous DHR and two (18.2%) patients had both simultaneous and sequential DHRs: one patient had immediate immune response to isoniazid and delayed immune response to pyrazinamide and cycloserine in the first (simultaneous form) and second (sequential form) oral provocation tests, respectively, while the other patient had delayed immune response to rifampin and ethambutol and immediate immune response to moxifloxacin in the first (simultaneous form), and second (sequential form) oral provocation tests, respectively. The median interval between initiation of TB treatment and the drug challenge test was 67 days, which was not significantly different between the two groups. There were no between-group differences in the interval between

initiation of TB treatment and the occurrence of the first sign of DHR and the interval between discontinuation of TB treatment and the oral provocation test.

## Comparisons of causative drugs and laboratory findings between single and multiple DHR groups

The causative drugs for DHRs in both groups are shown in Table 2. The median number of anti-TB drugs in the multiple DHR group was 2.0 (IQR 2.0–3.0; P < 0.000) (Table 3). Rifampin was the most common causative drug in the single and multiple DHR groups (n = 8 [47.1%] vs. (n = 9 [52.9%], respectively). Isoniazid, pyrazinamide, and moxifloxacin were significantly more common in the multiple DHR group than in the single DHR group.

In both groups, the peak values for eosinophil counts, eosinophil percentage, ALT levels, and AST levels were significantly elevated compared with the initial values for each variable (P values < 0.005 for all variables). However, there were no between-group differences in the initial and peak values for each variable (Table 4).

## Treatment outcomes between the single and multiple DHR group

The total duration of anti-TB treatment was not significantly different between the single and multiple DHR groups (365.5 days vs. 375.0 days; P = 0.610) (Table 5). The success rate was higher in the single DHR group than in the multiple DHR group; however, the difference was not statistically significant (94.1% vs. 81.8%; P = 0.543).

## Comparisons of clinical characteristics, oral provocation test results, and treatment outcomes according to the clinical manifestations of DHRs

The most common clinical manifestation of DHRs in all patients was maculopapular exanthema (MPE) (n = 23, 82.1%) (Table 6). Compared to MPE, anaphylaxis and DRESS syndrome tended to occur in younger patients; however, there was no statistically significant difference (p = 0.276).

Table 2. Anti-TB drugs causing DHRs.

| Anti-TB drugs | Numbers of patients |
|---|---|
| **Single DHRs** | **17** |
| RIF | 8 |
| EMB | 8 |
| PZA | 1 |
| **Multiple DHRs** | **11** |
| RIF + INH | 1 |
| RIF + EMB | 2 |
| RIF + PZA | 1 |
| EMB + PZA | 1 |
| RIF + PZA + moxifloxacin | 1 |
| RIF + EMB + moxifloxacin | 2 |
| INH + PZA + cycloserine | 1 |
| INH + RIF + EMB + PZA | 2 |

Data are presented as number. DHR, drug hypersensitivity reaction; TB, tuberculosis; INH, isoniazid; RIF, rifampin; EMB, ethambutol; PZA, pyrazinamide

**Table 3. Comparison of causative drugs in the single and multiple DHR groups.**

| Variables | Total (n = 28) | Single DHR (n = 17) | Multiple DHR (n = 11) | P value |
|---|---|---|---|---|
| Number of causative drug(s) | 1.0 (1.0–2.0) | 1 | 3.0 (2.0–3.0) | <0.000 |
| Isoniazid | 4 (14.3%) | 0 (0%) | 4 (36.4%) | 0.016 |
| Rifampin | 17 (60.7%) | 8 (47.1%) | 9 (81.8%) | 0.115 |
| Ethambutol | 15 (53.6%) | 8 (47.1%) | 7 (63.6%) | 0.460 |
| Pyrazinamide | 7 (25.0%) | 1 (5.9%) | 6 (54.5%) | 0.007 |
| Moxifloxacin | 3 (10.7%) | 0 (0%) | 3 (27.3%) | 0.050 |
| Cycloserine | 1 (3.6%) | 0 (0%) | 1 (9.1%) | 0.393 |

Data are presented as number (percentage) or median (interquartile range). DHR, drug hypersensitivity reaction.

## Discussion

In this study, we compared the characteristics of multiple and single DHRs associated with anti-TB drugs. To the best of our knowledge, this single center study is the first to compare single and multiple DHRs associated with anti-TB drugs. Rifampin was the most common causative drug for DHRs in the single and multiple DHR group. The treatment success rate in the multiple DHR group appeared to be lower than that in the single DHR group, although the difference was not statistically significant.

The oral provocation test is the gold standard test for the diagnosis of DHRs and identification of the causative drugs [7]; nonetheless, the diagnosis of DHRs can be supported by skin

**Table 4. Comparison of laboratory findings between the single and multiple DHR groups.**

| Variables | Total | Single DHR | Multiple DHR | P |
|---|---|---|---|---|
| WBC (×10$^3$/mm$^3$) | | | | |
| Initial | 5.5 (4.2–6.9) | 5.4 (4.4–7.1) | 5.6 (3.8–6.8) | 0.517 |
| Peak | 6.4 (3.9–8.2) | 6.4 (3.9–8.1) | 4.8 (3.3–8.4) | 0.578 |
| Lymphocyte (×10$^3$/mm$^3$) | | | | |
| Initial | 1.2 (0.9–2.0) | 1.2 (0.9–1.8) | 1.3 (0.8–2.1) | 0.677 |
| Peak | 1.0 (0.6–1.6) | 0.6 (0.9–1.3) | 1.0 (0.4–2.0) | 0.711 |
| Eosinophil (×10$^3$/mm$^3$) | | | | |
| Initial | 100 (37–200)[b] | 180 (40–200)[b] | 100 (20–110)[b] | 0.357 |
| Peak | 900 (200–1900)[b] | 600 (200–1400)[b] | 910 (300–2470)[b] | 0.611 |
| Eosinophil (%) | | | | |
| Initial | 1.9 (0.7–3.2)[b] | 2.2 (0.4–3.4)[b] | 1.8 (0.9–2.5)[b] | 0.890 |
| Peak | 12.3 (4.3–25.9)[b] | 11.5 (3.5–25.9)[b] | 16.0 (3.5–37.0)[b] | 0.487 |
| AST (U/L) [a] | | | | |
| Initial | 21.5 (19.0–31.7)[b] | 21.0 (18.5–29.7)[b] | 22.5 (19.0–41.2)[b] | 0.771 |
| Peak | 43.5 (34.7–97.0)[b] | 43.0 (35.0–60.0)[b] | 111.5 (33.0–443.2)[b] | 0.120 |
| ALT (U/L) [a] | | | | |
| Initial | 14.5 (9.7–24.5)[b] | 16.0 (10.0–29.0)[b] | 13.5 (9.2–17.2)[b] | 0.341 |
| Peak | 45.0 (25.0–102.5)[b] | 38.5 (27.2–58.5)[b] | 85.0 (19.5–227.5)[b] | 0.140 |

[a]Patients with immediate-type DHRs were excluded (one patient each in the single and multiple DHR groups).

[b] P value < 0.05 in comparisons between initial and peak values within groups

Data are presented as median (interquartile range). DHR, drug hypersensitivity reaction; WBC, white blood cell; AST, aspartate aminotransferase; ALT, alanine aminotransferase

**Table 5. Treatment outcomes between the single and multiple DHR groups.**

| Variables | Total n = 28 | Single DHR n = 17 | Multiple DHR n = 11 | P value |
|---|---|---|---|---|
| Total duration of anti-TB treatment, days | 370.0 (276.0–457.0) | 365.5 (264.5–470.5) | 375.0 (288.0–457.0) | 0.610 |
| Success rate | 25 (89.3%) | 16 (94.1%) | 9 (81.8%) | 0.543 |
| Cure | 10 (35.7%) | 6 (35.3%) | 4 (36.4%) | 1.000 |
| Complete | 15 (53.6%) | 10 (58.8%) | 5 (45.5%) | 0.700 |
| Unfavorable outcomes | 3 (10.7%) | 1 (5.9%) | 2 (18.2%) | 0.543 |
| Loss to follow-up | 2 (7.1%) | 1 (5.9%) | 1 (9.1%) | 1.000 |
| Death | 1 (3.6%) | 0 (0%) | 1 (9.1%) | 0.393 |

Data are presented as number (percentage) or median (interquartile range). DHR, drug hypersensitivity reaction; TB, tuberculosis

tests, such as skin prick, intradermal, or patch tests, based on the underlying mechanisms (immediate or delayed reactions) [7, 28, 29]. Multiple DHR is a hypersensitivity reaction to two or more chemically distinct drugs. It was classified two forms based on simultaneous and sequential sensitization of drugs leading to multiple DHR; a simultaneous form is multiple sensitizations at the same time, and a sequential form is multiple sensitization in a temporal sequence [23]. In all patients, except two patients, all DHRs to the causative drugs developed simultaneously. As TB treatment requires a combination of many anti-TB drugs, simultaneous

**Table 6. Clinical characteristics, oral provocation tests results, and treatment outcomes according to the clinical manifestations of DHRs.**

| | MPE (n = 22) | Anaphylaxis (n = 3) | DRESS (n = 3) |
|---|---|---|---|
| Age, years (IQR) | 60.5 (39.7–74.2) | 70.0 (23.0–74.0) | 44.0 (18.0–52.0) |
| Female sex | 14 | 3 | 1 |
| Initial symptoms of DHR | Skin rash (n = 22) | Skin rash (n = 2) | Skin rash (n = 3) |
| | Pruritus (n = 12) | Pruritus (n = 1) | Fever (n = 2) |
| | Fever (n = 5) | Fever (n = 2) | Eosinophilia (n = 3) |
| | Eosinophilia (n = 4) | Eosinophilia (n = 1) | |
| | | Syncope (n = 1) | |
| | | Hypotension (n = 1) | |
| Symptoms of DHR to OPT | Pruritus (n = 21) | Pruritus (n = 2) | Pruritus (n = 1) |
| | Rash (n = 17) | Rash (n = 2) | Rash (n = 3) |
| | Fever (n = 6) | Fever (n = 1) | Fever (n = 1) |
| | Hypotension (n = 1) | Urticaria (n = 2) | |
| | Dyspnea (n = 1) | Angioedema (n = 1) | |
| | | Hypotension (n = 1) | |
| | | Dyspnea (n = 2) | |
| Culprit drugs | INH (n = 3) | RIF (n = 3) | INH (n = 1) |
| | RIF (n = 11) | Mfx (n = 1) | RIF (n = 3) |
| | EMB (n = 14) | PZA (n = 2) | EMB (n = 1) |
| | PZA (n = 5) | | |
| | Mfx (n = 1) | | |
| Culprit drug and symptoms of further DHR after OPT | Cs (n = 1); pruritus and rash | - | Mfx (n = 1); fever and rash |
| Treatment success | 19 | 3 | 3 |
| Unfavorable outcomes | 3 | 0 | 0 |

Data are presented as number. MPE, maculopapular exanthema; DRESS, drug reaction with eosinophilia and systemic symptoms; IQR, interquartile range; OPT, oral provocation test; DHR, drug hypersensitivity reaction; INH, isoniazid; RIF, rifampin; EMB, ethambutol; PZA, pyrazinamide; Cs, cycloserine; Mfx, moxifloxacin.

drug sensitization may occur. Further, two patients for whom the causative drugs were identified after the first oral provocation test underwent the second oral provocation test because of recurrent DHRs associated with modified anti-TB drugs. In one patient, the causative drugs were identified as rifampin and ethambutol in the first oral provocation test. However, after the addition of moxifloxacin to the treatment regimen of isoniazid and pyrazinamide without the causative drugs, DHR recurred after 30 days of drug administration, and the second oral provocation test revealed that moxifloxacin was the culprit drug. Hence, the treatment regimen was changed to isoniazid, pyrazinamide, and levofloxacin. The patient then completed the treatment without further recurrence of DHRs. For the other patient, the causative drugs were identified as isoniazid and pyrazinamide in the first oral provocation test and cycloserine in the second test. The patient received ethambutol, moxifloxacin, prothionamide, and para-aminosalicylic acid. Although there was no further recurrence of DHR, the patient died 7 months later owing to aspiration pneumonia. These two patients had developed simultaneous and sequential multiple DHRs. As TB treatment regimens consist of many drugs, simultaneous multiple DHR were identified in most cases. However, combined combined simultaneous and sequential DHR to anti-TB drugs may occur because previous DHR may be a risk factor for the occurrence of subsequent DHRs [14].

The clinical symptoms and laboratory findings were not different between the single and multiple DHR groups. However, eosinophil counts, eosinophil percentage, and AST and ALT levels were significantly increased during DHRs compared to the baseline values in both groups. The frequency of DRESS syndrome appeared to be higher in the multiple DHR group than in the single DHR group (2 [18.2%] patients vs. 1 [5.9%] patients), however, the difference was not statistically significant (P = 0.543). The rRegiSCAR score in all patients with DRESS syndrome was > 4. The frequency of DRESS syndrome in patients with multiple DHRs in this study (18%) was lower than in previous studies (36%–57%) [30, 31] possibly owing to different culprit drugs and patient characteristics, including age, sex, and race [27, 30–33]. Although it was possible that patients who developed DRESS syndrome during the first DHR may have experienced DHRs with secondary or tertiary DRESS syndrome [14], there was no recurrence of DRESS syndrome in the present study.

Several studies evaluating the association between causative anti-TB drugs and DHRs have reported varying results. Pyrazinamide was the most common anti-TB drug that caused DHRs in previous studies [34–36]. However, rifampin and ethambutol were also reported as the most common agents associated with cutaneous ADRs, but they were not reported as multiple DHR causative drugs [15, 37]. In our study, consistent with the previous study, rifampin was the most common causative drug in the single DHR drug. In addition, it was also the most common causative drug in the multiple DHR group. Furthermore, the combination of rifampin-ethambutol was most common in the multiple DHR group. Further studies are needed to determine whether rifampicin and ethambutol influence each other in DHRs.

ADRs induced by anti-TB drugs can have a considerable impact on patient quality of life [8], affecting treatment adherence and increasing the risk of treatment failure and relapse [11, 12]. In this study, the treatment success rate in the multiple DHR group seemed to be lower than that in the single DHR group; however, there was no statistically significant difference. Moerover, the treatment success rate (84.9%) in all patients with DHRs in this study was lower than that in the previous cohort study in our institute and that in the multicenter cohort study in South Korea [3, 4]. The discontinuation of effective anti-TB drugs, including rifampin, due to DHRs contributed to the low treatment success rate in this study.

There are several limitations in this study. First, this was a single-center retrospective study, which limits the generalizability of our study findings. However, drug challenge tests were used to diagnose DHRs in all enrolled patients and the number of enrolled patients in this

study was larger than that in previous studies. Second, we only enrolled patients who underwent the oral provocation test, regardless of whether they underwent the skin prick test or patch test. This increases the likelihood of a selection bias in this study. Third, in our institute, desensitization of culprit drugs in patients with DHRs was not used, and second-line drugs which could have weak activity against to *Mycobacterium tuberculosis* were used for treating TB. Therefore the treatment outcome in patients with DHR could be influenced by using less effective second-line anti-TB drugs.

## Conclusion

DHRs to anti-TB drugs were common in all patients, and rifampin was the most common causative drug associated with multiple DHRs. The treatment outcomes appeared to be poorer in patients with TB with multiple DHRs than that in those with single DHRs.

## Supporting information

**S1 Table. Summary of enrolled subjects.**
(DOCX)

## Author Contributions

**Conceptualization:** Young-Il Koh, Yong-Soo Kwon.

**Data curation:** Hong-Joon Shin, Jin-Sun Chang, Min-Suk Kim, Bo-Gun Koh, Ha-Young Park.

**Formal analysis:** Tae-Ok Kim, Chul-Kyu Park, In-Jae Oh, Yu-Il Kim.

**Funding acquisition:** Yong-Soo Kwon.

**Investigation:** Hong-Joon Shin, Jin-Sun Chang, Sung-Chul Lim, Young-Chul Kim.

**Methodology:** Young-Il Koh, Yong-Soo Kwon.

**Project administration:** Tae-Ok Kim, Chul-Kyu Park, In-Jae Oh, Yu-Il Kim.

**Resources:** Hong-Joon Shin, Jin-Sun Chang, Young-Il Koh, Yong-Soo Kwon.

**Supervision:** Sung-Chul Lim, Young-Chul Kim, Young-Il Koh, Yong-Soo Kwon.

**Validation:** Min-Suk Kim.

**Visualization:** Jin-Sun Chang, Bo-Gun Koh, Ha-Young Park.

**Writing – original draft:** Hong-Joon Shin, Jin-Sun Chang.

**Writing – review & editing:** Hong-Joon Shin, Jin-Sun Chang, Min-Suk Kim, Bo-Gun Koh, Ha-Young Park, Tae-Ok Kim, Chul-Kyu Park, In-Jae Oh, Yu-Il Kim, Sung-Chul Lim, Young-Chul Kim, Young-Il Koh, Yong-Soo Kwon.

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
