## [Decision Letter · Decision Letter 0]

4 Aug 2020

PONE-D-20-16505

Hypersensitivity Reactions to Multiple Anti-tuberculosis Drugs

PLOS ONE

Dear Dr. Yong-Soo Kwon,

We have received two detailed reviews of your paper and after careful consideration, we feel that it has merit but does not fully meet PLOS ONE’s publication criteria as it currently stands. Therefore, we invite you to submit a revised version of the manuscript that addresses the points raised during the review process.

We look forward to receiving your revised manuscript.

Kind regards,

Walter R. Taylor

Academic Editor

PLOS ONE

2. Thank you for stating in the text of your manuscript " Informed consent was waived owing to the retrospective nature of the study, and patient information was anonymized and de-identified prior to analysis". Please also add this information to your ethics statement in the online submission form.

Reviewers' comments:

Reviewer's Responses to Questions

**Comments to the Author**

1. Is the manuscript technically sound, and do the data support the conclusions?

Reviewer #1: Partly

Reviewer #2: Partly

2. Has the statistical analysis been performed appropriately and rigorously? 

Reviewer #1: I Don't Know

Reviewer #2: Yes

3. Have the authors made all data underlying the findings in their manuscript fully available?

Reviewer #1: No

Reviewer #2: No

4. Is the manuscript presented in an intelligible fashion and written in standard English?

Reviewer #1: No

Reviewer #2: Yes

5. Review Comments to the Author

Reviewer #1: This manuscript addresses the issue of multiple drug hypersensitivity in the treatment of TBC. While there are published case reports, this seems to be the first retrospective chart analysis of a larger collective of patients.

major comment:

Methods

line 75 ff: Am I correct in understanding that you performed oral provocation tests in all patients with clinical suspicion of drug allergy without prior skin testing or performing other diagnostic tests? This would be not in line with current recommendations and guidelines on the management of drug allergy and could be highly dangerous in DRESS. Please specify. It seems, as if some patients at least did receive skin tests (discussion second but last paragraph). If so, please present the results.

Results:

line 19/20 and table 1: it would be more interesting to discuss the time between start of treatment, first signs of hypersensitivity reaction, time between stop of treatment and oral challenge. Also: at the time of the challenges: where the patients treated with alternative/second line treatment for the TB? Any information on the inflammatory state (e.g.CRP, BSR, neutrophils, IL-6....) at the time

table 1:

- calculating percentages in single cases of two collectives that overlap is misleading, e.g. in line "Previous treatment history": 1/28 patients in the whole collective and 1/17 patients in the subgroup result in different percentages, but in the end, this is only one patient. I suggest to omit the percentages from the small subgroups.

- skin rash: could you be more precise on the type and severity of clinical manifestation? macula-papular exanthema, AGEP, SDRIFE, FDE.......SJS/TEN

- table 2/3/4: please separate immediate type/anaphylaxis and delayed type reactions or alternatively, omit the few immediate cases from the table 4. One would not expect any elevation in liver parameters etc. in anaphylaxis.

Discussion line 15: what do you mean with “safely underwent the oral provocation tests”? Since you were able to confirm the clinical diagnosis of DHR by means of OPT those patients obviously reacted to the challenge, an at least unpleasant, if not to say potentially dangerous reaction that at least some of them could have been spared had you done a complete allergy workup beforehand.

- A separate table of the outcome of the diagnostic drug challenge as compared to the initial reaction would be useful with regard to the type and severity of the initial reaction, the type and severity of the reaction to the oral provocation and the further course of the treatment and potential further reactions etc. Did any of the patients with initial DRESS syndrome develop signs of systemic reaction upon oral provocation or did they “only” develop cutaneous signs?

Discussion line 40 “There were discrepancies….” Could you further speculate on the reason for this discrepancy based on a comparison between your patient collective and the other ones published? Also, based on the mode of action of rifampicin and the other drugs in question?

Discussion line 53: crossreactivity between rifampicin and ethambutol? Is there evidence for that? If yes, based on what? The chemical structure and mode of action are quite different.

Discussion line 58: please speculate on the reason for the reduced treatment success rate

minor comments:

Introduction line 40 ff: please be more precise along the lines of the ICON-paper you have cited.. A DHR is not "similar" to drug allergy and DHR do usually not occur at any time or after "many" days

Introduction line 57: omit with

Methods:

lines 79 ff: please specify the setting for immediate and delayed drug allergy: drug challenges in the hospital, at home, how long and how are the patients monitored

table 1: allergic disease: do you mean atopic disease? if yes, please change, if no, please specify

Results (Comparisons of causative drugs and laboratory findings between single and multiple

drug hypersensitivity groups, line 11 ff and table 4: this analysis is completely missing in the method section.

Discussion in general, esp. last two paragraphs and figure 1: proofreading by a native speaker is needed, there are a number of grammatical errors.

Reviewer #2: This is a useful manuscript that documents outcomes of an oral challenge of patients who developed hypersensitivity reactions to first-line TB drugs. There are some improvements that can be made to this manuscript that would add significantly to its value in a field where there is limited data.

Major Queries

1. Line 92-95 – was DRESS the only reaction studied. If so then this should be clarified in the introduction. If not the criteria for diagnosing the others should be included

2. How as anaphylaxis diagnosed?

3. Table 1 Are pruritus and fever always markers of hypersensitivity? Or are these part of a specific phenotype of ADR? It would be better if the authors classified/stratified the cases based on phenotype (i.e DRESS;anaphylaxis; sjs/ten; maculopapular etc.

4. The authors need to elaborate more on these.

5. How were positive reaction to oral challenge determined? Were there ant clinical features or only laboratory features? I suggest a table that includes every patient and the clinical features of a reaction, its timing etc. This will help others to identify the clinical features of this rechallenge reactions.

6. It would be useful to detail the interval between initial ADR and challenge reaction for each patient and compare this to those who did not develop a challenge reaction. It is well established that the closer the challenge is performed to the index ADR the higher is the likelihood of a challenge reaction. (40 vs 28)

7. Table 4. Does the initial event refer to the index drug reaction and which lab findings are reflected here. An average or the highest value during the event?

8. Line 110 and table 4 – what were the unfavorable outcomes. Initially defined as including all but in the table they are separated.

9. Lines 17 to 26 – the details of these reactions are useful addition to existing literature and even if added as supplementary material.

Minor Comments

1. Line 49 – loss not lost

2. Line 52 – “Previously, we reported that peripheral eosinophilia during anti-TB therapy was common (17.8%) and that cutaneous ADRs were common in these patients” ADR is common in all patients or those with eosinophilia?

3. Line 74 - anti-tuberculous change to antiTB for consistency

4. Line 76 – “starting with the drug with the lowest probability of hypersensitivity,” how was this established? Please provide reference

5. Line 77 – was there any rationale/references for the protocol used? Pls elaborate

6. Line 88 – “A simultaneous multiple DHR was the simultaneous occurrence of drug hypersensitivity to different administered drugs” is this the same definition as used by Gex-Collet et al?

7. Line 95 – meant to read “exclusion of other potential causes”

8. Line 97-100 is a repetition

9. Line 5 – should read flowchart

6. PLOS authors have the option to publish the peer review history of their article (what does this mean?). If published, this will include your full peer review and any attached files.

Reviewer #1: No

Reviewer #2: **Yes: **Rannakoe Lehloenya

---

## [Author Response · Author response to Decision Letter 0]

2 Dec 2020

PONE-D-20-16505

Hypersensitivity Reactions to Multiple Anti-tuberculosis Drugs

Walter R. Taylor

Academic Editor

PLOS ONE

Dear Dr. Taylor, 

Thank you for your letter dated August 26, 2020. We appreciate you and the reviewers for reviewing our manuscript entitled “Article Title: Hypersensitivity Reactions to Multiple Anti-tuberculosis Drugs” and offering helpful suggestions. We are submitting a revised manuscript that addresses the concerns that the reviewers raised. A detailed point-by-point response follows below. 

We look forward to any additional comments or questions concerning this paper and hope that the revised manuscript is now acceptable for publication in PLOS ONE.

Sincerely,

Yong-Soo Kwon, on behalf of all the authors.

Associate professor

Department of Internal Medicine

Chonnam National University Hospital

Gwangju, Korea

E-mail: yskwon@jnu.ac.kr

Reviewer #1: This manuscript addresses the issue of multiple drug hypersensitivity in the treatment of TBC. While there are published case reports, this seems to be the first retrospective chart analysis of a larger collective of patients.

1. Major comment:

Q1. Methods

Line 75 ff: Am I correct in understanding that you performed oral provocation tests in all patients with clinical suspicion of drug allergy without prior skin testing or performing other diagnostic tests? This would be not in line with current recommendations and guidelines on the management of drug allergy and could be highly dangerous in DRESS. Please specify. It seems, as if some patients at least did receive skin tests (discussion second but last paragraph). If so, please present the results.

Response: We agree with your comment that the oral provocation test is not recommended in patients with DRESS syndrome. However, since first-line anti-TB drugs are highly effective and few alternative effective anti-TB drugs are available, it is necessary to identify the culprit drug and use the remaining first-line anti-TB drugs. Therefore, we carefully performed the oral provocation tests using escalating doses as described in the manuscript. Of the 28 enrolled patients, two and eight patients underwent skin prick and patch tests, respectively, before the oral provocation test. One patient underwent both skin prick and patch tests before the oral provocation test. We have added this information in the Results section. Moreover, in this study, two patients with DRESS syndrome underwent patch tests before the oral provocation test; the patch test results were negative (Supplement table 1)

Q2. Results:

Q2-1) line 19/20 and table 1: it would be more interesting to discuss the time between start of treatment, first signs of hypersensitivity reaction, time between stop of treatment and oral challenge.

Response: Thank you for your comment. We have added the time interval between initiation of TB treatment and the occurrence of the first sign of DHR and the time interval between discontinuation of TB treatment and the oral provocation test in Table 1. We have also mentioned these findings in the Results section as follows: “There were no differences in the interval between initiation of TB treatment and the occurrence of the first sign of DHR and the interval between discontinuation of TB treatment and the oral provocation test between the two groups.”

Q2-2. Also: at the time of the challenges: where the patients treated with alternative/second line treatment for the TB? Any information on the inflammatory state (e.g.CRP, BSR, neutrophils, IL-6....) at the time

Response: Of the 28 patients, one patient underwent an oral provocation test while changing to second-line anti-TB drugs. The remaining 27 patients underwent oral provocation tests after discontinuing all anti-TB drugs. After identifying the causative drugs for DHRs, 19 and nine patients were administered modified anti-TB regimens during hospitalization and in outpatient clinics, respectively. We have added this in the Results section (Line 139). We have also added data on WBC count, neutrophil count, and CRP level, which serve as inflammatory markers on the oral provocation test day, in Table 1.

Q3. table 1:

Q3-1. calculating percentages in single cases of two collectives that overlap is misleading, e.g. in line "Previous treatment history": 1/28 patients in the whole collective and 1/17 patients in the subgroup result in different percentages, but in the end, this is only one patient. I suggest to omit the percentages from the small subgroups.

Response: We have omitted the percentages of the subgroups in Table 1 as per your recommendation.

Q3-2. skin rash: could you be more precise on the type and severity of clinical manifestation? macula-papular exanthema, AGEP, SDRIFE, FDE.......SJS/TEN

Response: We classified DHRs according to clinical manifestations, including MPE, anaphylaxis, DRESS syndrome, and SJS/TEN. We presented this information in Table 6. 

Q3-3. table 2/3/4: please separate immediate type/anaphylaxis and delayed type reactions or alternatively, omit the few immediate cases from the table 4. One would not expect any elevation in liver parameters etc. in anaphylaxis.

Response: We agree with your comment on the unlikelihood of liver enzyme level elevation in patients with immediate-type hypersensitivity. To avoid complex tables presenting immediate- and delayed-type reactions with single and multiple DHRs, we have omitted two patients with immediate-type reactions, including one with single DHRs and one with multiple DHRs, in the comparison of AST and ALT levels in Table 4. 

Q4. Discussion

Q4-1. line 15: what do you mean with “safely underwent the oral provocation tests”? Since you were able to confirm the clinical diagnosis of DHR by means of OPT those patients obviously reacted to the challenge, an at least unpleasant, if not to say potentially dangerous reaction that at least some of them could have been spared had you done a complete allergy workup beforehand.

Response: Thank you for your comment. We agree with your comment on the potential risk of OPT in patients with DHRs. Hence, we have deleted the sentence. 

Q4-2. A separate table of the outcome of the diagnostic drug challenge as compared to the initial reaction would be useful with regard to the type and severity of the initial reaction, the type and severity of the reaction to the oral provocation and the further course of the treatment and potential further reactions etc. Did any of the patients with initial DRESS syndrome develop signs of systemic reaction upon oral provocation or did they “only” develop cutaneous signs?

Response: We have added a separate table for the clinical characteristics, oral provocation test results, and treatment outcomes according to the phenotypes of DHR.

Table 6. Clinical characteristics, oral provocation tests results, and treatment outcomes according to the clinical manifestations of DHRs

 MPE (n = 22) Anaphylaxis (n = 3) DRESS (n = 3)

Age, years (IQR) 60.5 (39.7–74.2) 70.0 (23.0–74.0) 44.0 (18.0–52.0)

Female sex 14 3 1 

Symptoms of DHR to OPT Pruritus (n = 21)

Rash (n = 17)

Fever (n = 6)

Hypotension (n = 1)

Dyspnea (n = 1) Pruritus (n = 2)

Rash (n = 2)

Fever (n = 1)

Urticaria (n = 2)

Angioedema (n = 1)

Hypotension (n = 1)

Dyspnea (n = 2) Pruritus (n = 1)

Rash (n = 3)

Fever (n = 1)

Culprit drugs INH (n = 3)

RIF (n = 11)

EMB (n = 14)

PZA (n = 5)

Mfx (n = 1) RIF (n = 3)

Mfx (n = 1)

PZA (n = 2) INH (n = 1)

RIF (n = 3)

EMB (n = 1)

Culprit drug and symptoms of further DHR after OPT Cs (n = 1); pruritus and rash - Mfx (n = 1); fever and rash

Treatment success 19 3 3 

Unfavorable outcomes 3 0 0 

Data are presented as number. MPE, maculopapular exanthema; DRESS, drug reaction with eosinophilia and systemic symptoms; IQR, interquartile range; OPT, oral provocation test; DHR, drug hypersensitivity reaction; INH, isoniazid; RIF, rifampin; EMB, ethambutol; PZA, pyrazinamide; Cs, cycloserine; Mfx, moxifloxacin

Q4-3. line 40 “There were discrepancies….” Could you further speculate on the reason for this discrepancy based on a comparison between your patient collective and the other ones published? Also, based on the mode of action of rifampicin and the other drugs in question?

Response: There could be differences in the frequency of DRESS syndrome according to different drugs used and patient age, sex, and race (Ann Allergy Asthma Immunol 123 (2019) 483-487). We have added this in the discussion section as follow: “possibly owing to different culprit drugs and patient characteristics, including age, sex, and race.”

Q4-4. line 53: cross reactivity between rifampicin and ethambutol? Is there evidence for that? If yes, based on what? The chemical structure and mode of action are quite different.

Response: To the best of our knowledge, no study has evaluated whether rifampicin and ethambutol have cross-reactivity. To clarify the meaning, we have modified the sentence as follows: “Further studies are needed to determine whether rifampicin and ethambutol influence each other in DHRs.”

Q4-5. line 58: please speculate on the reason for the reduced treatment success rate

Response: Thank you for your comment. We have added the following reason in the revised manuscript: “The discontinuation of effective anti-TB drugs, including rifampin, due to DHRs contributed to the low treatment success rate in this study.”

2. Minor comments:

Q1. Introduction 

Q1-1. line 40 ff: please be more precise along the lines of the ICON-paper you have cited.. A DHR is not "similar" to drug allergy and DHR do usually not occur at any time or after "many" days

Response: Thank you for your comments. We made the following modification in the revised manuscript: “A drug hypersensitivity reaction (DHR) is an important adverse drug reaction (ADR) defined as “objectively reproducible symptoms or signs initiated by exposure to a defined stimulus at a dose tolerated by normal persons” by the World Health Organization (WHO). A drug reaction with demonstrated immunological mechanisms, either antibody or cell mediated, is referred to as drug allergy. DHRs can be classified as immediate or non-immediate DHRs depending on the onset time after drug exposure. Immediate DHRs typically occur within 1–6 h after drug exposure, whereas non-immediate DHRs commonly occur at any time after 1 h of drug administration” in the Introduction section (Line 41). 

Q1-2. line 57: omit with

Response: Thank you for your comment. We have omitted it.

Q2. Methods

Q2-1. lines 79 ff: please specify the setting for immediate and delayed drug allergy: drug challenges in the hospital, at home, how long and how are the patients monitored

Response: All patients who underwent the oral provocation test were hospitalized, and the test was performed and monitored by an allergologist. We have added this in the Methods section (Line 78). 

Q2-2. table 1: allergic disease: do you mean atopic disease? if yes, please change, if no, please specify

Response: Thank you for your comment. We have revised “allergic disease” to “atopic disease.”

Q3. Results (Comparisons of causative drugs and laboratory findings between single and multiple drug hypersensitivity groups, line 11 ff and table 4: this analysis is completely missing in the method section.

Response: Thank you for your comment. We have added the following in the Methods section: “Initial laboratory parameters at the start of anti-TB treatment and event laboratory parameters with the highest value when the patients had DHRs were collected. The laboratory parameters included white blood cell, lymphocyte, and eosinophil counts and percentages as well as aspartate aminotransferase (AST) and alanine aminotransferase (ALT) levels.” (Line 116)

Q4. Discussion in general, esp. last two paragraphs and figure 1: proofreading by a native speaker is needed, there are a number of grammatical errors.

Response: Thank you for your comment. Our manuscript has been proofread by a native English speaker. 

Reviewer #2: This is a useful manuscript that documents outcomes of an oral challenge of patients who developed hypersensitivity reactions to first-line TB drugs. There are some improvements that can be made to this manuscript that would add significantly to its value in a field where there is limited data.

Major Queries

Q1. Line 92-95 – was DRESS the only reaction studied. If so then this should be clarified in the introduction. If not the criteria for diagnosing the others should be included

Response: Thank you for your comment. We have added the criteria for diagnosing anaphylaxis and SJS/TEN in the Methods section as follows (Line 102): 

“SJS and TEN represent a spectrum of the most severe DHRs, wherein large areas of the epidermis become rapidly necrotic, resulting in skin detachment. SJS and TEN involve <10% and >30% of total body surface area, respectively.” 

Q2. How as anaphylaxis diagnosed?

Response: We have added information regarding the diagnosis of anaphylaxis in the Methods section as follows (Line 100): “An anaphyaxis was defined as an acute onset of an illness involving the skin, mucosal tissue, or both in more than one body system after exposure to a trigger.”

Q3. Table 1 Are pruritus and fever always markers of hypersensitivity? Or are these part of a specific phenotype of ADR? It would be better if the authors classified/stratified the cases based on phenotype (i.e DRESS;anaphylaxis; sjs/ten; maculopapular etc.) The authors need to elaborate more on these.

Response: Thank you for your comment. We have modified Table 1 as per your comment. We have classified DHR in terms of the phenotype (MPE, anaphylaxis, DRESS syndrome, and SJS/TEN), which is presented in Table 1.

Q5. How were positive reactions to oral challenge determined? Were there any clinical features or only laboratory features? I suggest a table that includes every patient and the clinical features of a reaction, its timing etc. This will help others to identify the clinical features of this rechallenge reactions.

Response: We added the following details about the oral provocation test in the Methods section: “All patients were hospitalized for the oral provocation test when they had no symptoms related to DHRs; an allergologist and a trained nurse performed the test and monitored DHR-related symptoms and signs such as pruritus and skin rash. When the patient developed sign and symptoms of DHRs, the test was considered positive.” Further, we have added a table including all patients and their individual DHR-related characteristics, oral provocation test results, and DHR timing (Supplement 1). 

Q6. It would be useful to detail the interval between initial ADR and challenge reaction for each patient and compare this to those who did not develop a challenge reaction. It is well established that the closer the challenge is performed to the index ADR the higher is the likelihood of a challenge reaction. (40 vs 28)

Response: Thank you for your comment. We have added the interval between the occur acne of the first sign of DHR and the oral provocation test in Table 1. However, we could not compare these in patients with positive and negative oral provocation test results. Of the 12 patients excluded from the study (Figure 1), only one patient underwent the oral provocation test; however, the patient was excluded as the final diagnosis was changed to NTM pulmonary disease. Of the remaining 11 patients excluded from the study, six and five patients underwent the skin prick test and the patch test, respectively. 

Q7. Table 4. Does the initial event refer to the index drug reaction and which lab findings are reflected here. An average or the highest value during the event?

Response: Thank you for your comment. The initial laboratory parameters at the start of anti-TB treatment and event laboratory parameters with the highest value when the patients had the DHRs were collected. We added to this information in the Method section as follows: “We collected laboratory parameters related to anti-TB treatment initiation (initial values) and DHR occurrence (peak values). The laboratory parameters included white blood cell, lymphocyte, and eosinophil counts and percentages as well as aspartate aminotransferase (AST) and alanine aminotransferase (ALT) levels.”

Q8. Line 110 and table 4 – what were the unfavorable outcomes. Initially defined as including all but in the table they are separated.

Response: The definition of unfavorable outcome is stated in the Methods section as follows: “Treatment failure, death, and loss to follow-up were considered as unfavorable outcomes.” No patient in this study had treatment failure; therefore, we have omitted the number of patients with treatment failure in Table 5. 

Q9. Lines 17 to 26 – the details of these reactions are useful addition to existing literature and even if added as supplementary material.

Response: Thank you for your valuable comment. Unfortunately, we could not add it, because we could not find any reference to sequential multiple DHR associated with anti-TB drugs. 

Minor Comments

Q1. Line 49 – loss not lost

Response: We have changed the word as per your comment.

Q2. Line 52 – “Previously, we reported that peripheral eosinophilia during anti-TB therapy was common (17.8%) and that cutaneous ADRs were common in these patients” ADR is common in all patients or those with eosinophilia?

Response: Thank you for your comment. We made the following modification in the revised manuscript: “In our previous study, peripheral eosinophilia during anti-TB therapy was common (17.8%) and cutaneous ADRs were common in these patients with eosinophilia.”

Q3. Line 74 - anti-tuberculous change to anti-TB for consistency.

Response: Thank you for your comment. We have changed the word as per your comment.

Q4. Line 76 – “starting with the drug with the lowest probability of hypersensitivity,” how was this established? Please provide reference

Response: Thank you for your comment. The revised manuscript has been modified to clarify the meaning. “The oral provocation test was performed in the order of pyrazinamide, isoniazid, ethambutol, and rifampin. However, according to the judgment of the clinician, the order of anti-TB drugs for oral provocation test may be changed in the order of drugs with the lowest probability of hypersensitivity.”

Q5. Line 77 – was there any rationale/references for the protocol used? Pls elaborate

Response: The protocol used in this study was modified from the one we previously used in our hospital (Kim et al. Allergy Asthma Immunol Res. 2014 March;6(2):183-185). We have cited the reference in the revised manuscript. 

Q6. Line 88 – “A simultaneous multiple DHR was the simultaneous occurrence of drug hypersensitivity to different administered drugs” is this the same definition as used by Gex-Collet et al?

Response: We have modified the manuscript as follows: “Simultaneous and sequential multiple DHRs were defined as the occurrence of hypersensitivity to different administered drugs simultaneously and at time intervals, respectively.”

Q7. Line 95 – meant to read “exclusion of other potential causes”

Response: We have corrected the word as per your comment.

Q8. Line 97-100 is a repetition

Response: We have omitted the sentence.

Q9) Line 5 – should read flowchart

A9. We have revised “flowcahrt” to “flowchart.”

---

## [Editor Report · Decision Letter 1]

14 Dec 2020

PONE-D-20-16505R1

Hypersensitivity Reactions to Multiple Anti-tuberculosis Drugs

PLOS ONE

Dear Dr. Yong-Soo Kwon, 

Thank you for submitting your manuscript to PLOS ONE. After careful consideration, we feel that it has merit but does not fully meet PLOS ONE’s publication criteria as it currently stands. Therefore, we invite you to submit a revised version of the manuscript that addresses the points raised during the review process.

We look forward to receiving your revised manuscript.

Kind regards,

Walter R. Taylor

Academic Editor

PLOS ONE

Additional Editor Comments (if provided):

Dear Dr. Yong-Soo Kwon,

thankyou for resubmitting the revised paper.

You cover an important area and the paper is much improved.

However, there are some small changes I am requesting.

Please give the reader an idea of the risk of developing an adverse drug reaction when on standard anti TB treatment reported by others.

Do you know how many patients were treated for TB during the time of your retrospective study ? That would also give us an idea of risk.

In your previous study, you reported eosinophilia in patients with TB drug reactions. More detail is needed like the rash morphologies and the eosinophil counts.

Please add a few simple lines to explain to general physicians what a sequential and simultaneous drug reaction are.

There is a small error regarding the Wilcoxon Signed Rank test - it is used to assess paired (within group) differences.

Please tell the reader how you made a successful diagnosis of a positive rechallenged test. Table 6 would be improved by detailing the key symptoms and signs of the initial drug reaction and what happened when that individual was rechallenged. Therefore, you need to list every patient. Rank them by initial reaction starting with mpe. Please delete treatment outcomes as this has already been covered in Table 5.

Table 4. We need to see the significant p values within groups as well. As you seem to have the white cell data, a graph would be very nice showing the changes in total and white cell count differentials over time to cover also the oral provocation test (this is optional).

It is worth reminding clinicians in the Discussion that drug reactions can occurs months after starting treatment and referral to a specialist unit for drug rechallenged should be made quickly.

yours sincerely,

Walter Taylor.

---

## [Author Response · Author response to Decision Letter 1]

28 Dec 2020

PONE-D-20-16505

Hypersensitivity Reactions to Multiple Anti-tuberculosis Drugs

Walter R. Taylor

Academic Editor

PLOS ONE

Dear Dr. Taylor, 

Thank you for your letter dated August 26, 2020. We appreciate you for reviewing our manuscript entitled “Article Title: Hypersensitivity Reactions to Multiple Anti-tuberculosis Drugs” and offering helpful suggestions. We are submitting a revised manuscript that addresses the concerns that the reviewers raised. A detailed point-by-point response follows below. 

We look forward to any additional comments or questions concerning this paper and hope that the revised manuscript is now acceptable for publication in PLOS ONE.

Sincerely,

Yong-Soo Kwon, on behalf of all the authors.

Associate professor

Department of Internal Medicine

Chonnam National University Hospital

Gwangju, Korea

E-mail: yskwon@jnu.ac.kr

 

1. Please give the reader an idea of the risk of developing an adverse drug reaction when on standard anti TB treatment reported by others.

A) Thank you for your comment. We added the risk of developing an adverse drug reactions in the Introduction section as you suggested: “However, adverse drug reactions (ADRs) to anti-TB drugs can cause significant morbidity, leading the discontinuation of these effective anti-TB drugs and worse outcomes. The risk of developing an ADR to anti-TB drugs has been variously reported from 8 to 85% depending on the population the studies enrolled and the severity of ADRs the studies evaluated”

2. Do you know how many patients were treated for TB during the time of your retrospective study? That would also give us an idea of risk.

A) Thank you for your comment. As described in Figure 1, a total of 2,347 patients received anti-TB treatment during the study period. We added it in the Results section.

3. In your previous study, you reported eosinophilia in patients with TB drug reactions. More detail is needed like the rash morphologies and the eosinophil counts. 

A) Thank you for your comment. Since this is a retrospective study, we could not evaluate the rash morphologies, but we described the eosinophil counts related to anti-TB treatment initiation (initial values) and DHR occurrence (peak values) in Table 4. 

4. Please add a few simple lines to explain to general physicians what a sequential and simultaneous drug reaction are.

A) Thank you for your comment. We added the additional explanation for sequential and simultaneous DHR in the Discussion section as “Multiple DHR is a hypersensitivity reaction to two or more chemically distinct drugs. It was classified two forms based on simultaneous and sequential sensitization of drugs leading to multiple DHRs; a simultaneous form is multiple sensitizations at the same time, and a sequential form is multiple sensitization in a temporal sequence.”. 

5. There is a small error regarding the Wilcoxon Signed Rank test - it is used to assess paired (within group) differences.

A) Thank you for your valuable comment. We performed the Wilcoxon Signed Rank test to determine whether the peak lab was significantly elevated than the baseline lab (within group, paired test). We changed the description in the statistical analysis as “The Wilcoxon signed rank test was used to compare initial and peak laboratory variables within groups”

6. Please tell the reader how you made a successful diagnosis of a positive rechallenged test. Table 6 would be improved by detailing the key symptoms and signs of the initial drug reaction and what happened when that individual was rechallenged. Therefore, you need to list every patient. Rank them by initial reaction starting with MPE. Please delete treatment outcomes as this has already been covered in Table 5.

A) Thank you for your comment. We already described how can do the oral provocation test and when it was considered positive in the Method section. “All patients were hospitalized for the oral provocation test when they had no symptoms related to DHRs; an allergologist and a trained nurse performed the test and monitored DHR-related symptoms and signs such as pruritus and skin rash. When the patient developed sign and symptoms of DHRs, the test was considered positive.”

We added the initial symptoms of DHRs in table 6 and modified supplement table for describing symptoms and signs of every patient. Since the complexity of table, we listed symptoms and sings of every patient about of the initial drug reaction and what happened when that individual was rechallenged only in the supplement table. 

7. Table 4. We need to see the significant p values within groups as well. As you seem to have the white cell data, a graph would be very nice showing the changes in total and white cell count differentials over time to cover also the oral provocation test (this is optional).

A) Thank you for your comments. We added the significant p values in comparisons between initial and peak values within groups in Table 4. We did not add a graph because the WBC was not significantly different between groups and between initial and peak values within groups, and it has already been covered in table 4. 

8. It is worth reminding clinicians in the Discussion that drug reactions can occurs months after starting treatment and referral to a specialist unit for drug rechallenged should be made quickly.

A) Thank you for your comment. We added the additional comment to the Discussion section as your suggestion. “Clinicians should keep in mind that DHR can occur several months after the start of anti-TB treatment, and consider referral to a specialized unit to identify culprit drugs quickly.”.

---

## [Editor Report · Decision Letter 2]

18 Jan 2021

Hypersensitivity Reactions to Multiple Anti-tuberculosis Drugs

PONE-D-20-16505R2

Dear Dr. Yong-Soo Kwon,

We’re pleased to inform you that your manuscript has been judged scientifically suitable for publication and will be formally accepted for publication once it meets all outstanding technical requirements.

Kind regards,

Walter R. Taylor

Academic Editor

PLOS ONE
---

## [Editor Report · Acceptance letter]

26 Jan 2021

PONE-D-20-16505R2 

Hypersensitivity Reactions to Multiple Anti-tuberculosis Drugs 

Dear Dr. Kwon:

I'm pleased to inform you that your manuscript has been deemed suitable for publication in PLOS ONE. Congratulations! Your manuscript is now with our production department. 

Kind regards, 

on behalf of

Dr. Walter R. Taylor 

Academic Editor

PLOS ONE